# Neoadjuvant immunotherapy with nivolumab and ipilimumab induces major pathological responses in patients with head and neck squamous cell carcinoma

Joris L. Vos [1,18], Joris B. W. Elbers[2,18], Oscar Krijgsman[3], Joleen J. H. Traets[4,5], Xiaohang Qiao [5], Anne M. van der Leun [4], Yoni Lubeck[6], Iris M. Seignette[6], Laura A. Smit[6], Stefan M. Willems[7], Michiel W. M. van den Brekel [1,8], Richard Dirven[1,8], M. Baris Karakullukcu[1,8], Luc Karssemakers[1,8], W. Martin C. Klop[1,8], Peter J. F. M. Lohuis[1,8], Willem H. Schreuder[1,8], Ludi E. Smeele[1,8], Lilly-Ann van der Velden[1,8], I. Bing Tan[9], Suzanne Onderwater[1], Bas Jasperse[10], Wouter V. Vogel[11,12], Abrahim Al-Mamgani[12], Astrid Keijser[13], Vincent van der Noort[13], Annegien Broeks[14], Erik Hooijberg [6], Daniel S. Peeper [4,15], Ton N. Schumacher [4,15], Christian U. Blank [4,16], Jan Paul de Boer[16], John B. A. G. Haanen [4,16] & Charlotte L. Zuur[1,5,8,17 ✉]

Surgery for locoregionally advanced head and neck squamous cell carcinoma (HNSCC) results in 30–50% five-year overall survival. In IMCISION (NCT03003637), a non-randomized phase Ib/IIa trial, 32 HNSCC patients are treated with 2 doses (in weeks 1 and 3) of immune checkpoint blockade (ICB) using nivolumab (NIVO MONO, n = 6, phase Ib arm A) or nivo-lumab plus a single dose of ipilimumab (COMBO, n = 26, 6 in phase Ib arm B, and 20 in phase IIa) prior to surgery. Primary endpoints are feasibility to resect no later than week 6 (phase Ib) and primary tumor pathological response (phase IIa). Surgery is not delayed or suspended for any patient in phase Ib, meeting the primary endpoint. Grade 3–4 immune-related adverse events are seen in 2 of 6 (33%) NIVO MONO and 10 of 26 (38%) total COMBO patients. Pathological response, defined as the %-change in primary tumor viable tumor cell percentage from baseline biopsy to on-treatment resection, is evaluable in 17/20 phase IIa patients and 29/32 total trial patients (6/6 NIVO MONO, 23/26 COMBO). We observe a major pathological response (MPR, 90–100% response) in 35% of patients after COMBO ICB, both in phase IIa (6/17) and in the whole trial (8/23), meeting the phase IIa primary endpoint threshold of 10%. NIVO MONO's MPR rate is 17% (1/6). None of the MPR patients develop recurrent HSNCC during 24.0 months median postsurgical follow-up. FDG-PET-based total lesion glycolysis identifies MPR patients prior to surgery. A baseline AID/APOBEC-associated mutational profile and an on-treatment decrease in hypoxia RNA signature are observed in MPR patients. Our data indicate that neoadjuvant COMBO ICB is feasible and encouragingly efficacious in HNSCC.

---

A full list of author affiliations appears at the end of the paper.

Sixty-four percent of patients with head and neck squamous cell carcinoma (HNSCC) present with locoregionally advanced disease at diagnosis[1]. Depending on site and stage, these patients are treated with definitive (concurrent chemo) radiotherapy [(C)RT] or extensive surgery combined with free or pedicled flap reconstuction followed by adjuvant (C)RT. Still, 5-year overall survival after surgery is only 50%[2,3], with patients bearing HPV-negative tumors[4] and those undergoing salvage surgery for in-field residual or recurrent disease after (C)RT having a worse prognosis[5,6]. Furthermore, the majority of surgically treated HNSCC patients experience problems in areas as swallowing, speech or aesthetics[7]. These poor outcomes highlight the need for treatment options that improve survival or allow for de-intensification of the standard of care (SOC). Anti-PD-1 immune checkpoint blockade (ICB) has become standard first- and second-line palliative care for recurrent or metastatic (R/M) HNSCC[8–11], showing objective response rates of 13–17% in an unselected R/M-HNSCC population and 2-year overall survival rates of 17–27%[9,11,12]. Combination ICB targeting PD-1 and CTLA-4 using nivolumab and ipilimumab has shown higher efficacy than monotherapy in solid cancers, such as advanced melanoma and renal cell carcinoma[13,14]. However, a benefit of adding anti-CTLA-4 to anti-PD-1 ICB was not observed in patients with R/M-HNSCC[10,15].

In a curative setting, neoadjuvant ICB may be more effective than adjuvant ICB[16–18] across several tumor types[19–21]. Furthermore, this regimen allows for pathologic response evaluation and biomarker discovery[22]. However, neoadjuvant ICB could be challenging in HNSCC patients, as they typically suffer from multiple comorbidities, sequelae of previous HNSCC treatments in the context of field cancerization[23], and will face a highly intensive standard-of-care (SOC) treatment regimen after neoadjuvant therapy. In addition, the time frame available for neoadjuvant ICB in HNSCC in phase I/II trials is limited, as curative surgery later than 6 weeks after diagnosis is associated with worse outcome[24].

Evidence on neoadjuvant ICB in HPV-negative HNSCC is limited but suggests modest activity of anti-PD-1 monotherapy, with a major pathological response rate at the primary tumor site of 8–14%[25,26]. One study suggests combined anti-PD-1 and anti-CTLA-4 ICB may be more effective[25], yet this regimen's efficacy needs further evaluation. Here, we present a nonrandomized phase Ib/IIa trial (IMCISION, NCT03003637), in which we investigated the safety, feasibility and efficacy of neoadjuvant nivolumab monotherapy and nivolumab plus ipilimumab prior to SOC surgery in patients with HNSCC.

In IMCISION, 32 patients with T2–T4, N0–N3b, M0 primary or recurrent HNSCC, planned for surgery, are included. Patients received two courses of neoadjuvant ICB (weeks 1 and 3) prior to SOC surgery in week 5–6 with or without adjuvant (C)RT, according to institutional and national treatment guidelines. In phase Ib, 6 patients (arm A) were treated with nivolumab monotherapy (NIVO MONO, 240 mg flat dose) in weeks 1 and 3 as a safety run-in, and 6 patients (arm B) were treated with nivolumab (240 mg flat dose in weeks 1 and 3) plus ipilimumab (1 mg kg$^{-1}$, in week 1 only), further referred to as COMBO. The primary objective of phase Ib was safety and feasibility. Subsequently, a single-arm phase IIa extension cohort of 20 patients was treated with the COMBO ICB regimen. The primary objective of phase IIa was pathological efficacy at the time of surgery after neoadjuvant COMBO ICB. A response rate <10% was considered clinically irrelevant. Baseline tumor biopsies were taken prior to first ICB and on-treatment specimens were obtained at time of surgery. MR imaging and, if additional consent was given, 18F-fluorodeoxyglucose (FDG) PET/CT were performed at baseline and shortly prior to surgery (Fig. 1a).

Efficacy was assessed by pathological response (PR) evaluation of the primary tumor surgical specimen and RECIST measurements (v.1.1) of MR imaging. Multiplex immunofluorescence, immunohistochemistry and molecular translational research are presented. We show that neoadjuvant NIVO MONO or COMBO ICB can be administered safely, without jeopardizing surgical timelines, and leads to a major pathological response in a substantial minority of patients (17% and 35%, respectively).

## Results

**Patient characteristics.** Thirty-three patients were enrolled in IMCISION between February 28, 2017 and October 25, 2019 (Supplementary Fig. 1). One patient, scheduled for total glossectomy, experienced clinically evident tumor regression upon neoadjuvant COMBO ICB and refused surgery. This patient was treated off-study with nivolumab maintenance at another institute, and another patient was included. The patient was excluded from all analyses and remains alive with no evidence of disease at the time of writing, which has been 37 months since inclusion. The remaining 32 patients were predominantly males (63%), diagnosed with an oral cavity tumor (84%) and with a history of tobacco or alcohol use (84 and 81%, Table 1). One tumor was HPV-associated. Twenty-two patients (69%) were included with primary HNSCC: 18 as first, 2 second, 1 third, and 1 fifth primary HNSCC. Three patients (9%) had stage II, 9 (28%) had stage III and 10 (31%) had stage IVA-B disease. Ten patients (31%) had pretreated recurrent/residual HNSCC, of whom 6 had undergone prior RT, CRT or cetuximab-RT (hereafter referred to as salvaged patients, Supplementary Table 1). Salvaged patients were over-represented in the NIVO MONO (3 of 6, 50%) versus the COMBO cohort (3 of 26, 12%). Thirteen patients (41%) were included with clinically nodal-positive disease.

During IMCISION, two patients (pt21 and pt34) were found to be ineligible after enrollment. While pt21 was initially diagnosed with reflux esophagitis, this patient turned out to have a synchronous incurable esophageal carcinoma after completion of neoadjuvant treatment. For pt34, one cervical metastasis proved unresectable due to carotid artery encasement, retrospectively already present at baseline. Pt32, who was included eligibly with recurrent HNSCC after previous surgery with adjuvant RT, developed histologically confirmed, unresectable carcinomatous lymphangitis while on neoadjuvant treatment. Surgery was canceled in these three patients, who were subsequently treated with best supportive care. On-treatment biopsies were taken in these three patients (all enrolled in the phase IIa extension cohort). However, as PR assessment in a biopsy might not be representative for whole tumor response, these three patients were excluded from ICB pathological efficacy evaluation to maintain a uniform PR analysis. Twenty-nine patients (6 NIVO MONO, 23 COMBO) thus remained for definitive analysis. Survival analyses separated per pathological response category are reported from the time of surgery for these 29 patients. Overall survival is additionally reported for all 32 patients from the time of first ICB dose, where the three patients that did not undergo surgery are included based on their clinically assessed response: one with an assumed major pathological response (MPR, pt21) and two with no assumed pathological response (NPR, pt32, and pt34). Immune-related adverse events (irAEs) are reported for all 32 patients.

**Neoadjuvant ICB is safe and feasible prior to extensive surgery in HNSCC.** Thirty-one of 32 patients (97%) completed both courses of ICB; one patient (pt33, COMBO) refused the second cycle. SOC surgery was performed according to baseline tumor extent no later than week 6, a median of 27 days (IQR 2) after

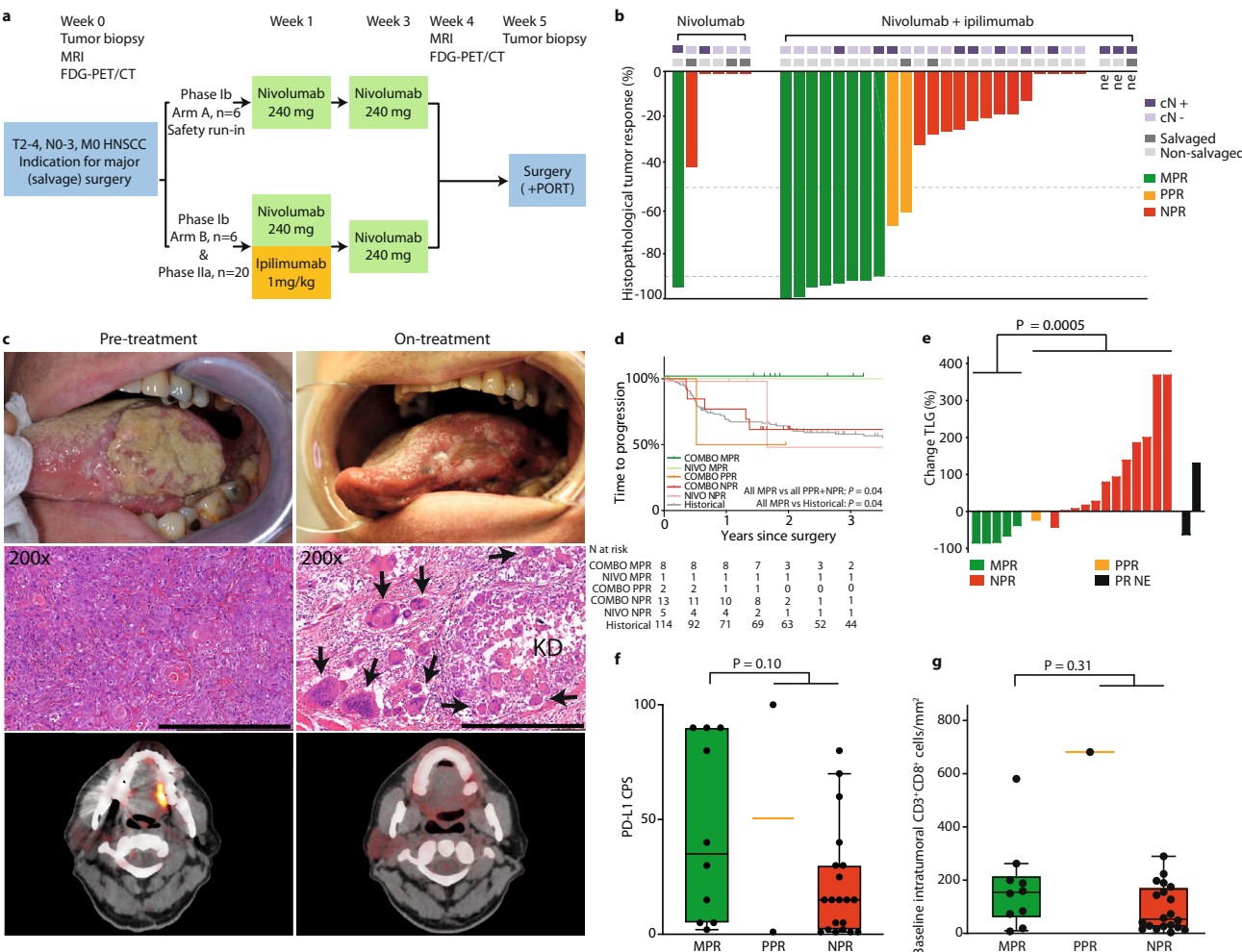

**Fig. 1 IMCISION trial design, pathological response waterfall plot, time-to-progression survival analysis, FDG-PET response evaluation, PD-L1 CPS, and tumor-infiltrating CD3+CD8+ T-cells. a** IMCISION phase Ib/IIa study design. Patients with primary or recurrent HNSCC, with an indication for major (salvage) head and neck surgery, were included in IMCISION. Patients underwent baseline primary tumor biopsy and several imaging studies. Phase Ib consisted of 12 patients (6 treated with nivolumab in arm A as a safety run-in, 6 with nivolumab + ipilimumab in arm B), phase IIa of 20 patients (all nivolumab + ipilimumab). Imaging was repeated in week 4, shortly prior to surgery. **b** Percentage pathological response (PR) at the primary tumor site from baseline biopsy to on-treatment surgical specimen in patients treated with nivolumab (17% MPR, left) and nivolumab + ipilimumab (35% MPR, right). The two dotted horizontal lines (at 50 and 90% PR) divide patients into three categories: 90–100% PR (major pathological response, MPR, green), 50–89% partial PR (PPR, yellow), and <50% or no PR (NPR, red). Three patients did not undergo surgery and are marked not evaluable ('ne'). Upper bars mark baseline clinical lymph node status and patients included with recurrent disease after previous (chemo / bio) radiotherapy ('salvaged'). Source data are provided as a Source Data file. **c** Clinical photography (upper panels), H&E-stained tumor sections (middle panels, scale bars measure 500 μm) and FDG-PET images (lower panels) of pt39 with a cT3N0 carcinoma of the left tongue border at baseline (left) and on-treatment, shortly prior to surgery (right). This patient achieved an MPR. The clinically evident reduction in tumor bulk is reflected on the H&E slide by the presence of keratinous debris (KD) and surrounding multinucleated giant cells (arrows). FDG-PET obtained shortly prior to surgery shows resolution of tracer accumulation at the tongue border. **d** Kaplan–Meier estimate of the time to progression of the 29 IMCISION patients, classified according to neoadjuvant treatment and primary tumor PR, and of 114 non-IMCISION patients from a comparable historical cohort that underwent major (salvage) surgery without neoadjuvant treatment. Comparisons are made using a two-sided log-rank test. None of the patients with an MPR upon neoadjuvant ICB (either NIVO or COMBO) suffered recurrent disease after 24.0 months median follow-up. **e** Waterfall plot showing the percentage change in primary tumor total lesion glycolysis (TLG) from baseline to on-treatment as measured by FDG-PET, stratified by PR. Black bars represent the TLG change in 2 patients without an evaluable pathological response ('PR NE'). One of the 'PR NE' patients was pt21, who had a TLG decrease and an evident clinical response (see Supplementary Fig. 2). An exact P-value was calculated using a two-sided Wilcoxon rank-sum test. Source data are provided as a Source Data file. **f** Baseline PD-L1 combined positive score (CPS) of primary tumors per PR category assessed per immunohistochemistry. An exact P-value was calculated using a two-sided Wilcoxon rank-sum test. N = all 32 baseline primary tumor samples (10 MPR, 2 PPR, 20 NPR). Source data are provided as a Source Data file. **g** Baseline intratumoral infiltration of CD3 + CD8 + T-cells assessed per digital analysis of multiplex-stained slides. An exact P-value was calculated using a two-sided Wilcoxon rank-sum test. N = 31 baseline primary tumor samples (10 MPR, 1 PPR, 20 NPR). Source data are provided as a Source Data file. For **f** and **g**, boxplots represent the median and 25th and 75th percentile, the whiskers extend from the hinge to the minimal and maximal data point but no further than 1.5× IQR. For translational research purposes, the three patients that did not undergo surgery were included for CPS and CD3 + CD8 + T-cell assessment and categorized according to their clinical ICB response: 1 as likely MPR and 2 as likely NPR.

**Table 1 Baseline patient and tumor characteristics of phase Ib, IIa, and total patients.**

| | Phase Ib patients (n = 12) | Phase IIa patients (n = 20) | Total patients (n = 32) |
|---|---|---|---|
| *Characteristic* | | | |
| Median age, years (range) | 63 (54–78) | 65 (22–76) | 65 (22–78) |
| Sex, *n (%)* | | | |
| Male | 8 (67) | 12 (60) | 20 (63) |
| Female | 4 (33) | 8 (40) | 12 (38) |
| Smoking history, *n (%)* | | | |
| Current or former | 12 (100) | 15 (75) | 27 (84) |
| Never | 0 | 5 (25) | 5 (16) |
| Alcohol use history, *n (%)* | | | |
| Current or former | 11 (92) | 15 (75) | 26 (81) |
| Never | 1 (8) | 5 (25) | 6 (19) |
| WHO performance status, *n (%)* | | | |
| 0 | 11 (92) | 13 (65) | 24 (75) |
| 1 | 1 (8) | 7 (35) | 8 (25) |
| Tumor site, *n (%)* | | | |
| Oral cavity | 8 (67) | 19 (95) | 27 (84) |
| Oropharynx | 3 (25) | 1 (5) | 4 (13) |
| Larynx | 1 (8) | 0 | 1 (3) |
| HPV status, *n (%)* | | | |
| Positive | 0 | 1 (5) | 1 (3) |
| Negative | 12 (100) | 19 (95) | 31 (97) |
| HNSCC status, *n (%)* | | | |
| Primary | 6 (50) | 16 (80) | 22 (69) |
| Recurrent | 6 (50) | 4 (20) | 10 (31) |
| Clinical T-stage, *n (%)* | | | |
| T2 | 2 (17) | 4 (20) | 6 (19) |
| T3 | 5 (42) | 10 (50) | 15 (47) |
| T4 | 5 (42) | 6 (30) | 11 (34) |
| Clinical N-stage, *n (%)* | | | |
| N0 | 7 (58) | 12 (60) | 19 (59) |
| N1 | 2 (17) | 4 (20) | 6 (19) |
| N2 | 3 (25) | 3 (15) | 6 (19) |
| N3 | 0 | 1 (5) | 1 (3) |
| Clinical disease stage (AJCC 8th ed), *n (%)* | | | |
| II | 1 (8) | 2 (10) | 3 (9) |
| III | 2 (17) | 7 (35) | 9 (28) |
| IV | 3 (25) | 7 (35) | 10 (31) |
| Recurrent | 6 (50) | 4 (20) | 10 (31) |

Percentages may not add up to 100 due to rounding.
*WHO* World Health Organization, *HPV* human papillomavirus, *HNSCC* head and neck squamous cell carcinoma, *AJCC* American Joint Committee on Cancer.

start of ICB. There was no delay in surgery due to irAEs (CTCAE v. 4.03), although progressive disease precluded surgery in one patient (pt32, COMBO). Immune-related AEs were observed in 4 NIVO MONO patients (67%, 95% CI: 22–96%) and 18 COMBO patients (69%, 95% CI: 48–86%, Supplementary Table 2). Two NIVO MONO (33%, 95% CI: 4–78%) and 10 COMBO patients (38%, 95% CI: 20–59%) developed grade 3-4 irAEs. Excluding asymptomatic laboratory abnormalities that spontaneously resolved, grade 3-4 irAEs were seen in 2 NIVO MONO (33%, 95% CI: 4–78%) and 3 COMBO patients (12%, 95% CI: 2–30%). No previously unknown or unexpected irAEs were observed. One patient developed grade 2 immune-related hepatitis (COMBO) and was treated with oral glucocorticoids during the neoadjuvant period. Three patients with grade 3 immune-related colitis (1 NIVO MONO, 2 COMBO) and 1 patient with grade 3 immune-related pericarditis (NIVO MONO) were treated with oral glucocorticoids, occurring a median of 45 days (range 33–78 days)

after start of ICB. Two of the three patients with grade 3 immune-related colitis (1 NIVO MONO, 1 COMBO) required second-line immune suppression with infliximab. All three cases of immune-related rash (all COMBO) were treated with topical glucocorticoids. Four patients (all COMBO) required chronic thyroid hormone replacement therapy after immune-related thyroiditis, all other grade 3-4 irAEs were resolved to grade 1 without sequelae.

Microscopically tumor-negative resection margins were achieved in all 6 NIVO MONO and 22 of 23 evaluable COMBO patients. Twenty-six patients (90%, 95% CI: 73–98%) experienced one or more grade ≥2 postsurgical complications according to Clavien-Dindo[27] (Supplementary Table 3). The median duration of hospital stay after surgery was 17 days (IQR 5) for NIVO MONO and 16 days (IQR 16) for COMBO patients. One patient (COMBO), who was discharged while on 80 mg prednisone per day due to grade 3 immune-related colitis, was readmitted 7 days later and remained hospitalized for 145 days with a septic wound infection of the fibula flap donor site.

According to national and institutional guidelines, 20 patients had an indication for adjuvant RT and 5 for adjuvant platinum-based CRT. Adjuvant radiotherapy was not performed in 10 of 20 patients: seven patients were previously irradiated in the head and neck area, two had delayed postoperative wound healing, and one patient refused. Four patients with an indication for adjuvant CRT refused systemic treatment in dread of toxicity and one was deemed unfit due to high age and comorbidities—all five were treated with RT only. RT-associated toxicity (CTCAE v. 4.03) in the 15 patients (10 with NPR, 4 with MPR, and 1 with a partial pathological response) who received adjuvant treatment was in line with experience in HNSCC (Supplementary Table 4).

**Neoadjuvant nivolumab + ipilimumab induces 35% major pathological responses at the primary tumor site of HNSCC.** Eight of 23 (35%, 95% CI: 16–57%) evaluable patients treated with COMBO and 1 of 6 (17%, 95% CI: 0–64%) with NIVO MONO had 90–100% PR at the primary tumor site (major pathological response, MPR). One patient (COMBO) had a pathological complete response and 1 (COMBO) only a 0.3 mm focus of residual viable tumor. Two of 23 COMBO patients (9%, 95% CI: 1–28%) had 50–89% PR (partial pathological response, PPR, Fig. 1b). Three COMBO patients with MPR, all with HNSCC of the lateral border of the tongue, reported clinical tumor shrinkage and increased tongue mobility during neoadjuvant treatment (Fig. 1c). Resection specimens of patients with MPR were characterized by areas of fibrosis, neovascularization, immune cell infiltration, residual keratinous debris with surrounding multinucleated giant cells, and aggregates of macrophages (Fig. 1c). None of the five pathologically evaluable salvaged patients, of whom three were in the NIVO MONO cohort, had an MPR (Fig. 1b). Of the three pathologically unevaluable COMBO patients, pt21 had a durable primary tumor response based on clinical evaluation, MR and FDG-PET, as well as an on-treatment biopsy showing keratinous debris, a dense immune infiltrate, neovascularization, fibrosis and only a small focus of residual viable tumor (Supplementary Fig. 2)[28]. The other two unevaluable COMBO patients (pt32 and pt34) had clinically progressive disease in week 5, and no evidence of ICB response in their on-treatment biopsy.

While 13 patients (11 COMBO, 2 NIVO MONO) had clinically tumor-positive cervical lymph nodes at baseline, pathological evaluation revealed nodal-positive disease in 16 patients (13 COMBO, 3 NIVO MONO). All four patients (3 COMBO, 1 NIVO MONO) with a primary tumor MPR and nodal metastases had residual tumor in their affected lymph nodes after ICB,

although two had evidence of a nodal ICB response (1 COMBO, 1 NIVO MONO). One PPR patient (COMBO) with lymph node metastases had residual nodal tumor without a nodal ICB response. Conversely, two patients with NPR in their primary tumor demonstrated evidence of a treatment effect in their lymph node metastases, achieving MPR in one or more affected lymph nodes (both COMBO, Supplementary Fig. 3).

Median follow-up time since surgery was 24.0 months (95% CI: 21.5–not attained): 46.5 months (95% CI: 45.1–not attained) for the NIVO MONO and 23.1 months (95% CI: 21.5–38.5) months for the COMBO cohort. The 29 pathologically evaluable IMCISION patients showed a similar time to progression (TTP, $P = 0.21$) and overall survival (OS, $P = 0.98$) compared to a historical cohort (characteristics in Supplementary Table 5) undergoing major (salvage) surgery at our hospital (Supplementary Fig. 4a, b). Neither the one patient with an MPR after NIVO MONO (95% CI: 0–98%) nor any of the 8 patients with an MPR after COMBO (95% CI: 0–37%) developed a tumor relapse, compared to 1 of the 5 (20%, 95% CI: 1–72%) non-MPR NIVO MONO and 6 of the 15 (40%, 95% CI: 16–68%) non-MPR COMBO patients. The NIVO MONO MPR patient and 6 of the 8 COMBO MPR patients are alive and disease-free, while 2 of the 8 (25%, 95% CI: 3–65%) COMBO MPR patients have died due to HNSCC-unrelated causes: one pneumonia and one diverticulitis. Four of five (80%, 95% CI: 28–99%) NIVO MONO non-MPR patients have died, of whom one due to HNSCC relapse and three due to HNSCC-unrelated causes: one case of impaired post-surgical wound healing, one pancreatic carcinoma, and one cardiovascular event. Of the 15 COMBO non-MPR patients, 6 have died (40%, 95% CI: 16–68%): five due to HNSCC relapse and one cardiovascular event. In case of an MPR upon neoadjuvant ICB (either NIVO MONO or COMBO), the Kaplan–Meier curve for TTP since surgery is significantly superior at a median follow-up of 24.0 months when compared to patients without MPR or to the historical cohort (both $P = 0.04$, Fig. 1d). Overall survival was not significantly superior for MPR compared to non-MPR patients after either NIVO MONO or COMBO at a median follow-up of 24.0 months, or the historical cohort ($P = 0.11$ and $P = 0.25$, respectively, Supplementary Fig. 4c).

Of the three pathologically unevaluable COMBO patients, two had died: pt32 of progressive HNSCC and pt21 due to esophageal carcinoma, although this patient died with no evidence of disease at the tongue carcinoma site (Supplementary Fig. 2). An additional overall survival analysis including all 32 patients who received neoadjuvant ICB (i.e., including pt32 and pt34 as COMBO NPR patients and pt21 as a COMBO MPR patient) since the first ICB dose (median follow-up 24.2 months, 95% CI 22.6–not attained) showed similar OS for patients with MPR and non-MPR patients ($P = 0.19$, Supplementary Fig. 4d).

**On-treatment FDG-PET identifies patients with early MPR to neoadjuvant ICB prior to surgery.** Radiological evaluation of neoadjuvant ICB response was performed using MR imaging at baseline and shortly prior to surgery. On-treatment MR was obtained a median of 3 days (IQR 0) prior to surgery. Paired MR was unavailable in six patients (four non-compliance, two unmeasurable disease), leaving 26 patients evaluable per MR-RECIST (v.1.1[29]). Of these 26, 7 had MPR, 2 had PPR, 15 had NPR, and 2 patients were pathologically unevaluable (pt21 and pt32). In line with reports in other tumor types[18,30,31], MR-RECIST underestimated the frequency and depth of the primary tumor pathological response. Of the 7 RECIST-evaluable patients with MPR after either NIVO MONO or COMBO, 2 showed a radiological partial response and 5 had stable disease. Both PPR patients had stable disease, and the 15 NPR patients showed

stable (10) or progressive disease (5). MR-RECIST's ability to detect MPR after NIVO MONO or COMBO ICB thus yielded a specificity of 100% (95% CI: 80–100%), but a sensitivity of 29% (95% CI: 4–71%) and an accuracy of 79% (95% CI: 58–93%), Supplementary Table 6). Of the two RECIST-evaluable but pathologically unevaluable patients, one had a partial response (pt21, Supplementary Fig. 2) and one had progressive disease (pt32).

Progressive disease (PD) per MR-RECIST was seen in 6 of 26 RECIST-evaluable patients (23%, 5 NPR patients and pt32). In 3 of these 6 RECIST PD-patients, disease was upstaged, which resulted in preclusion (1 patient) or alteration (1 patient) of the planned surgery in 2 patients, and to an indication for planned adjuvant therapy in 1 patient. Pt32's disease was upstaged from rT3N1M0 to rT3N2cM1, precluding surgery. In addition, pt32's tumor sum diameter increased from 44 to 69 mm (+57%), meeting the requirements of Matos et al.'s definition of hyperprogression[32]. One other patient with RECIST PD (pt37) was upstaged from T3N0M0 to T3N1M0 and the baseline surgical plan was altered from a hemiglossectomy to a subtotal glossectomy. Finally, one RECIST PD-patient (pt30) was upstaged from rT2N0M0 to rT4aN0M0 due to bone invasion that was not evident on baseline imaging. This patient's surgical plan remained unaltered, but the patient gained an indication for adjuvant RT. The remaining three RECIST PD-patients were neither upstaged, nor required alteration of their surgical or adjuvant therapy plan prior to surgery.

FDG-PET scans were not mandatory. Baseline total lesion glycolysis (TLG) of the primary tumor could be calculated from the FDG-PET scan for 26 pathologically evaluable patients (6 with MPR, 20 without MPR) and was lower for patients with MPR (median 11.8 g, IQR 76, 95% CI 10.6–303.4) compared to non-MPR patients (median 21.4 g, IQR 47, 95% CI 15.8–41.5), but not significantly ($P = 0.09$). If additional consent was given, an on-treatment scan was made a median of 3 days prior to surgery (IQR 0), and the percent change in TLG from baseline to on-treatment was available for 18 pathologically evaluable patients (5 with MPR, 13 without MPR). Patients with an MPR showed a strong median decrease in TLG after neoadjuvant ICB (−98% compared to baseline, IQR 38) compared to a median increase in patients without MPR (+92%, IQR 215), which was statistically significant ($P = 0.0005$, Fig. 1e). These data suggest that a TLG decrease after neoadjuvant ICB assessed by FDG-PET could be an early on-treatment imaging biomarker for MPR in HNSCC.

**Biomarker analyses of response to neoadjuvant ICB.** Biomarker results of the six patients treated with NIVO MONO are grouped together with the 26 treated with COMBO ICB. All 32 tumors were immunohistochemically microsatellite stable and HLA class 1-proficient (Supplementary Fig. 5a). Median PD-L1 combined positive score (CPS) at baseline was higher in patients with MPR (35, IQR 85) compared to patients without MPR (15, IQR 31), albeit not significantly ($P = 0.10$, Fig. 1f, Supplementary Fig. 5b). The primary tumor immune microenvironment of 31 baseline primary tumor samples was assessed per multiplex immunofluorescence and revealed a higher baseline median CD3 + CD8 + intratumoral T-cell density in patients with MPR (154, IQR 156) compared to patients without MPR (58, IQR 157), although this difference was not statistically significant ($P = 0.31$, Fig. 1g). Pairwise comparison of baseline and on-treatment multiplex-stained primary tumor samples could be performed in 26 patients (Supplementary Fig. 5c) and revealed a significant increase in intratumoral CD3 + CD8 + T-cell density in both MPR and non-MPR patients after ICB ($P = 0.04$ and $P = 0.0003$, respectively, Supplementary Fig. 5d).

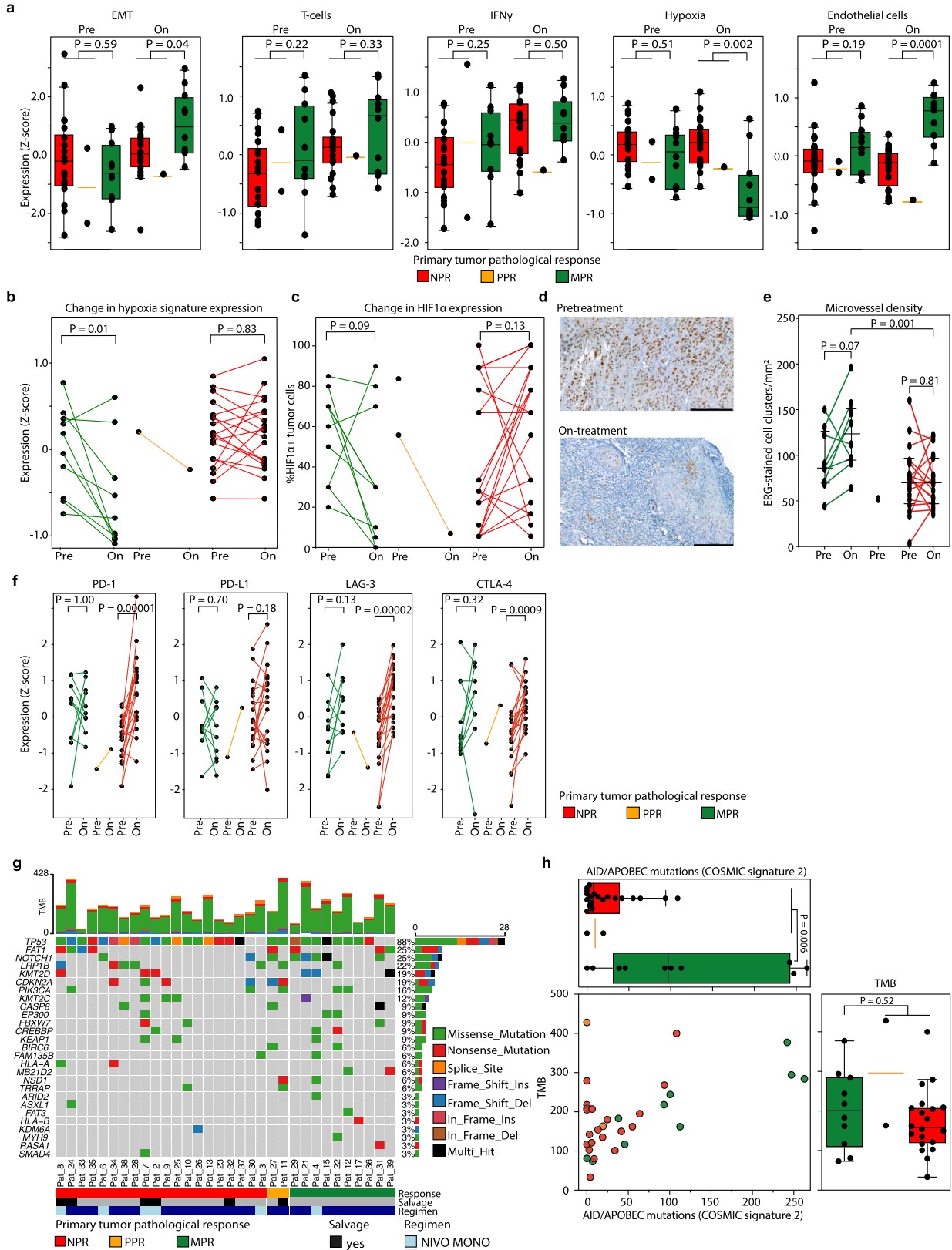

RNA sequencing could be performed on all 32 baseline and 30 on-treatment primary tumor biopsies. Based on geneset enrichment analysis, the epithelial to mesenchymal transition signature was enriched in baseline tumors of patients without MPR (Supplementary Fig. 6a), although its presence was insufficient to predict ICB response (Fig. 2a). Baseline and on-treatment IFNγ

and T-cell signature presence (Z-score) were not significantly higher in patients with an MPR (Fig. 2a, Supplementary Fig. 6b–d).

While hypoxia is known to unfavorably influence the outcome of HNSCC patients treated with radiotherapy[33], baseline primary tumor hypoxia-associated gene expression in IMCISION did not predict ICB response (Fig. 2a). However, on-treatment biopsies of

**Fig. 2 RNA and whole-exome sequencing. a** Baseline and on-treatment epithelial to mesenchymal transition (EMT)[68], T-cell[66], IFNγ[65], endothelial cell[66], and tumor hypoxia[33] expression signatures per pathologic response (PR) category assessed by RNA sequencing (RNAseq). Exact P-values were calculated using a two-sided Wilcoxon rank-sum test. Baseline $N =$ all 32 primary tumor samples (10 MPR, 2 PPR, 20 NPR), on-treatment $N = 30$ primary tumor samples (10 MPR, 1 PPR, 19 NPR). Source data are provided as a Source Data file. **b** Change in hypoxia gene expression in paired baseline and on-treatment primary tumor samples. An exact P-value was calculated using a two-sided Wilcoxon signed rank test. Baseline $N =$ all 32 primary tumor samples (10 MPR, 2 PPR, 20 NPR), on-treatment $N = 30$ primary tumor samples (10 MPR, 1 PPR, 19 NPR). Source data are provided as a Source Data file. **c** Change in the percentage of tumor cells that express hypoxia-inducible factor 1α (HIF-1α) in paired baseline and on-treatment primary tumors, measured by immunohistochemistry. Two MPR patients without analyzable, residual tumor after ICB are included with value '0%'. An exact P-value was calculated using a two-sided Wilcoxon signed rank test. Baseline $N =$ all 32 primary tumor samples (10 MPR, 2 PPR, 20 NPR), on-treatment $N = 30$ primary tumor samples (10 MPR, 1 PPR, 19 NPR). Source data are provided as a Source Data file. **d** HIF-1α-stained primary tumor slides of a patient with primary tumor MPR at baseline (top) and on-treatment (bottom). Black bars measure 200 μm. **e** Microvessel density (MVD) in available pre- and on-treatment primary tumor samples. Comparisons between pre- and on-treatment samples of the same patient are made using a two-sided Wilcoxon signed rank test. The comparison between the median MVD of on-treatment MPR and NPR samples is made using a two-sided Wilcoxon rank-sum test. All P-values are exact. Baseline $N = 29$ (9 MPR, 1 PPR, 19 NPR), on-treatment $N = 29$ (10 MPR, 0 PPR, 19 NPR). Source data are provided as a Source Data file. **f** Change in PD-1, PD-L1, LAG-3, and CTLA-4 signature expression in baseline and corresponding on-treatment primary tumor samples per PR category. An exact P-value was calculated using a two-sided Wilcoxon signed rank test. Baseline $N =$ all 32 primary tumor samples (10 MPR, 2 PPR, 20 NPR), on-treatment $N = 30$ primary tumor samples (10 MPR, 1 PPR, 19 NPR). Source data are provided as a Source Data file. **g** Oncoplot showing mutations as assessed by whole-exome sequencing (WES) of baseline primary tumor samples. Baseline $N =$ all 32 primary tumor samples (10 MPR, 2 PPR, 20 NPR); a column represents a patient. Top bar chart represents tumor mutational burden (TMB). Percentages listed right represent the proportion of samples harboring a mutation in the gene listed left. Bottom bars show PR, salvage status and ICB regimen. Source data are provided as a Source Data file. **h** Top plot: number of COSMIC[36] signature 2 (AID / APOBEC)-associated mutations per PR category. Right plot: total TMB per ICB response category. The dot plot shows ICB response and the contribution of AID / APOBEC-associated mutations to the TMB per individual sample. Exact P-values were calculated using a two-sided Wilcoxon rank-sum test. Baseline $N =$ all 32 primary tumor samples (10 MPR, 2 PPR, 20 NPR). Source data are provided as a Source Data file. For translational purposes, the three patients that did not undergo surgery were categorized according to their clinical ICB response in this figure: 1 as likely MPR and 2 likely NPR. For **a** and **h**, boxplots represent the median and 25th and 75th percentile, the whiskers extend from the hinge to the minimal and maximal data point but no further than 1.5× IQR.

MPR tumor samples showed significantly lower hypoxia gene expression when compared to non-MPR samples ($P = 0.002$, Fig. 2a, Supplementary Fig. 6b, e). Moreover, in a paired analysis of baseline and corresponding on-treatment samples, a significant decrease of hypoxia-related gene expression was observed in MPR biopsies, while this decrease was absent in non-MPR biopsies ($P = 0.01$, Fig. 2b). On-treatment hypoxia-inducible factor 1α (HIF-1α) staining correlated with hypoxia gene expression ($R = 0.62$, $P = 0.0002$, Supplementary Fig. 6f) and tended to decrease from baseline to on-treatment in MPR samples, albeit not significantly ($P = 0.09$, Fig. 2c, d). Microvessel density (MVD) could be assessed with immunohistochemistry for endothelial marker ERG in 29 on-treatment tumor samples (10 MPR, 19 NPR) and was significantly higher in MPR samples when compared to non-MPR samples ($P = 0.001$, Fig. 2e). Neovascularization has previously been noted as a histopathological characteristic of ICB response[34] and could be of interest in the context of our finding of low hypoxia and high endothelial cell signature expression in on-treatment MPR samples. Furthermore, high baseline and on-treatment tumor hypoxia gene expression negatively correlated with T-cell, B-cell, CD8 + T-cell, and endothelial cell signatures, and was positively correlated with neutrophil gene expression (Supplementary Fig. 6g). Finally, immune checkpoint signatures for PD-1, LAG-3, and CTLA-4 were significantly upregulated after neoadjuvant ICB in samples without MPR (Fig. 2f).

Tumor mutational burden (TMB) was assessed by whole-exome sequencing of baseline samples. The median TMB of the cohort was 172 mutations (IQR 120). Median TMB of MPR samples (201 mutations, IQR 179) compared to non-MPR samples (163 mutations, IQR 91) did not differ significantly ($P = 0.52$, Fig. 2g, h). Similar to previous reports[35], TP53 was mutated in a majority of samples (88%). FAT1 and NOTCH1 mutations were observed in 25% of the samples (Fig. 2g). No specific enrichment of gene mutations was identified in relation to ICB response. The most common COSMIC[36] (catalogue of somatic mutations in cancer) mutational signature found in all baseline samples was signature 1, marked by specific C > T mutations (Supplementary Fig. 7). In baseline primary tumor samples, COSMIC signature 2, thought to arise from cytidine deaminase activity of AID / APOBEC, was significantly enriched in MPR patients compared to patients without MPR ($P = 0.006$, Fig. 2h).

Presumed biomarkers PD-L1 CPS, intratumoral CD3 + CD8 + T-cell infiltration, and TMB (Supplementary Fig. 8a) or the number of baseline COSMIC signature 2 (AID / APOBEC)-associated mutations (Supplementary Fig. 8b) were analyzed in combination for their ability to predict response to ICB as shown in a bubble plot, illustrating an overrepresentation of MPR patients in the upper right quadrants. Pt32, who could not undergo surgery, had a baseline tumor characterized by a high CPS (40), but low TMB (80, trial median 172), low baseline IFNγ gene expression (Z-score −0.92, trial median −0.26) and low intratumoral CD8 + T-cells (41.7, trial median 83.8 cells/mm$^2$).

## Discussion

The IMCISION trial has shown that therapy with nivolumab or nivolumab plus ipilimumab prior to extensive surgery for patients with HNSCC is safe and does not delay standard-of-care. Still, grade 3–4 irAEs were seen in 33% of patients after NIVO MONO and 38% after COMBO ICB, underlining the necessity to closely monitor patients for the occurrence of irAEs in the neoadjuvant and postoperative phase. Two cycles of neoadjuvant COMBO ICB induced a major pathological response at the primary tumor site in 35% of patients with resectable HNSCC in a 4-week time frame, which is higher than the radiological objective response rate to anti-PD-1-based ICB observed in R/M-HNSCC (13–18%)[8–11] and complements reports in other solid tumor types[18,19,30,31,37,38]. Similar to previous reports in HNSCC[26,39] and lung cancer[40], ICB response was discordant between primary HNSCC and its lymph node metastases. Whether this discordance persists over time or turns out to be a delayed ICB response[41] should be addressed in future studies. TTP and OS of IMCISION patients were not significantly different from survival rates observed in a historical cohort of patients treated surgically

at our hospital, though low patient numbers and lack of a control group in IMCISION limit the interpretability of these results. Longer follow-up and larger cohorts in controlled trials are needed to investigate whether neoadjuvant ICB conveys a durable survival benefit in the general HNSCC population. In HNSCC, pathologic staging drives the indication for adjuvant therapy. In IMCISION, the decision to administer adjuvant (chemo)radiotherapy was based on both pretreatment imaging and physical examination as well as pathological staging of the surgical specimen. Adjuvant treatment was not altered based on a patient's response to ICB. However, it may be that any potential pathological indication for adjuvant (chemo)radiotherapy was cleared by neoadjuvant ICB in MPR patients, leading to relative adjuvant undertreatment. Still, none of the 9 MPR patients treated with either NIVO MONO or COMBO ICB in IMCISION developed a tumor relapse after a median follow-up of 2 years, irrespective of the discordant lymph nodal response and potential relative adjuvant undertreatment of the MPR group. Even if an unselected HNSCC population does not benefit from neoadjuvant ICB in terms of overall survival, we propose that these findings, together with the relatively high MPR rate especially after COMBO ICB, are encouraging and have opened the door for potential future trials investigating the possibilities to postpone or de-escalate extensive and mutilating surgery in patients with a likely MPR. Such an approach is further supported anecdotally by the durable response observed in pt21, in whom surgery was omitted. Response-driven treatment adaptation is currently under investigation in advanced melanoma patients[42].

The primary tumor MPR rate in IMCISION after COMBO ICB (35%) was higher than the 14% reported after neoadjuvant anti-PD-1 monotherapy in a recent phase II trial by Uppaluri et al. (calculated from supplementary table 4)[26]. Uppaluri et al. administered one cycle of pembrolizumab to 36 HPV-negative HNSCC patients (66% oral cavity) and performed surgery 13–22 days later. In another phase II trial, Schoenfeld et al.[25] treated 29 patients with oral cavity carcinoma with two cycles of nivolumab ($n = 14$) or nivolumab plus ipilimumab (given in week 1 only, $n = 15$) and performed surgery 3–7 days after the second cycle. Schoenfeld et al. report an MPR in 4 of 29 (14%) patients; 1 of 14 (7%) NIVO MONO, and 3 of 15 (20%) COMBO patients. IMCISION was not designed to compare NIVO MONO to COMBO, and there is currently no evidence supporting the addition of ipilimumab to nivolumab in the neoadjuvant HNSCC setting, which requires further study. Still, the greater MPR rate in IMCISION after COMBO ICB, which was also observed by Schoenfeld et al.[25], suggests that patients with HPV-negative HNSCC may benefit from neoadjuvant COMBO ICB. IMCISION's two neoadjuvant ICB cycles and the relatively long time frame between the second cycle and surgery may further explain its relatively high MPR rate. Mouse experiments suggest that an optimal window for neoadjuvant ICB and the timing of surgery exists, as both a relatively long (causing T-cell dysfunction due to prolonged tumor antigen stimulation) and a relatively short time frame (interrupting effective T-cell expansion due to tumor antigen and intratumoral T-cell removal) abrogate the benefit of ICB[43]. In the CIAO-trial, investigating neoadjuvant ICB in patients with mainly HPV-positive oropharyngeal HNSCC, Ferrarotto et al. observed a primary tumor MPR in only 2 of 25 evaluable patients (8%) after two cycles of durvalumab (anti-PD-1, $n = 13$) or durvalumab with tremelimumab (anti-CTLA-4, $n = 12$), one in each treatment arm[39]. Surgery was performed 52–72 days after the start of ICB in CIAO. Given the observation that none of the salvaged patients in IMCISION, overrepresented in the NIVO MONO cohort, achieved MPR, the inclusion of 31% salvaged patients in CIAO may have negatively influenced the MPR rate in their study.

All FDG-PET-evaluable patients with MPR demonstrated a decrease in total lesion glycolysis from baseline to on-treatment in IMCISION, as was also shown in a report on neoadjuvant ICB in NSCLC within a similar time frame[44]. While the value of FDG-PET-based ICB response assessment in HNSCC needs validation, it may be a readily available surrogate marker for histopathological response to select MPR patients for postponement of surgery in future trials. Progressive disease was seen in 6 of 26 (23%) MR-RECIST-evaluable patients, which precluded surgery in one patient. While this patient had hyperprogression according to the definition proposed by Matos et al.[32], the absence of data on the natural course of this patient's disease without neoadjuvant ICB treatment makes it hard to say whether this patient indeed had ICB-associated hyperprogression. RECIST PD further led to significant expansion of the planned surgical plan in one and to escalation of the proposed adjuvant therapy plan in another patient. The occurrence of progressive disease in the neoadjuvant phase highlights the need for biomarker-based selection of HNSCC patients who are likely to benefit, and timely surgery in those unlikely to benefit from neoadjuvant ICB.

The molecular investigations of IMCISION revealed a significant correlation between an increased pretreatment AID/APOBEC tumor mutational signature and MPR upon neoadjuvant NIVO MONO or COMBO ICB. AID and APOBEC are enzymes able to induce clustered C > T mutations and play a role in innate immunity, but have also been implied in cancer development[45,46]. An APOBEC mutational pattern has been positively correlated with immune infiltration in HNSCC[47,48], and with immunotherapy response in a TCGA analysis of multiple solid tumors, including HNSCC[49]. This result suggests that, aside total TMB, the type and etiology of mutations are similarly important for (neoadjuvant) ICB response in HNSCC. The utility of a high AID/APOBEC mutational signature as a baseline biomarker to select or stratify for HNSCC patients that will respond early upon ICB should be further assessed.

Tumor hypoxia is a well-established negative prognostic factor in HNSCC[33], suppressing effective antitumor immunity[50] and conveying nonresponse and poor OS after ICB in the R/M setting[51]. Interestingly, baseline primary tumor hypoxia gene expression did not significantly differ between response categories in IMCISION. Nevertheless, we did observe a negative correlation between baseline and on-treatment hypoxia and immune cell infiltration, with the exception of neutrophils. In addition, on-treatment samples of patients with MPR showed a low hypoxia signature expression, which was accompanied by a decrease in HIF-1α-positive tumor cells in MPR after ICB. Whether a decrease in hypoxia could be an early on-treatment biomarker for MPR in HNSCC requires further study.

Taken together, IMCISION shows that nivolumab and nivolumab plus ipilimumab prior to surgery are effective and safe regimens for patients with resectable and predominantly HPV-negative HNSCC, resulting in a primary tumor MPR in 35% of COMBO and 17% of NIVO MONO patients. Moreover, none of the patients with an MPR at the primary tumor site after either NIVO MONO or COMBO ICB has developed a tumor relapse at a median of 2 years postsurgical follow-up, though without an OS benefit in this cohort. IMCISION has further identified an AID/APOBEC-associated mutational profile as a potential future biomarker for the baseline selection of patients who are more likely to achieve MPR after neoadjuvant NIVO MONO or COMBO. Finally, a reduction in total lesion glycolysis assessed by FDG-PET scans and a decrease in hypoxia gene expression are two possible on-treatment biomarkers that may identify patients with MPR early upon ICB. IMCISION provides strong rationales for future neoadjuvant ICB trials in HNSCC patients, ultimately

aiming at improving their survival and de-escalating their intensive and mutilating standard of care.

## Methods

**Patients**. Patients 18 years of age or older with histologically confirmed T2–T4, N0–N3b, M0 primary or recurrent, resectable HNSCC of the oral cavity, oropharynx, hypopharynx, or larynx were eligible for inclusion in IMCISION (NCT03003637), and were recruited between February 28, 2017 and October 25, 2019. First primary HNSCC was defined as a patient's first occurrence of HNSCC. New primary HNSCC (second primary, third primary and so on) was defined as the reoccurrence of HNSCC more than 5 years after treatment of previous HNSCC, or the reoccurrence of HNSCC in a different anatomical subsite[52]. Any reoccurrence of HNSCC within 5 years after treatment of HNSCC in the same anatomical subsite was considered recurrent HNSCC. When the trial commenced, only patients with T3–T4 were eligible for inclusion; this criterion was expanded in March 2019 to allow for the inclusion of patients with bulky T2 tumors. All patients had an indication for major curative (salvage) surgery. Patients enrolled prior to January 2018 were staged according to AJCC 7th edition; for all subsequent patients, AJCC 8th edition was used. In the present manuscript, all patients' TNM and disease stage are reported per AJCC 8th edition. All patients had a World Health Organization (WHO) Performance Score (PS) of 0 or 1. Main exclusion criteria were the presence of autoimmune disease, human immunodeficiency virus or hepatitis B/C infection; prior immunotherapy targeting CTLA-4, PD-1, or PD-L1 and use of immunosuppressive medication.

**Trial design, interventions, treatments, and endpoints**. Thirty-three patients were included in this nonrandomized phase Ib/IIa trial carried out at the Netherlands Cancer Institute (NKI). Patients were to receive two courses of neoadjuvant ICB (weeks 1 and 3) prior to standard of care (SOC) surgery in week 5-6. Twelve patients were included in phase Ib, designed as a double 3 + 3 trial. Phase Ib primary endpoints were safety and feasibility. Study treatment would be deemed unsafe and not feasible if more than one of six patients treated with either nivolumab monotherapy (NIVO MONO, phase Ib arm A) or nivolumab + ipilimumab dual ICB (COMBO, phase Ib arm B) would not be able to undergo SOC surgery in ultimately week 6 due to immune-related adverse events (irAEs). Immune-related AEs and other AEs were scored in terms of the Common Terminology Criteria for Adverse Events (CTCAE) version 4.03 for up to 100 days after the last ICB dose. By order of accrual, the first 3 phase Ib patients were treated with nivolumab 240 mg flat dose in weeks 1 and 3 prior to surgery in week 5-6 (NIVO MONO, arm A). After being demonstrably safe and feasible according to the primary outcome, three additional NIVO MONO patients were included in arm A. When the primary safety endpoint was met in 5 or more these first 6 patients part of the IMCISION trial's safety run-in, neoadjuvant dual ICB (COMBO) was administered to the next 3 patients by order of accrual (arm B), consisting of nivolumab 240 mg flat dose + ipilimumab 1 mg kg$^{-1}$ (week 1) followed by nivolumab 240 mg (week 3). After again establishing this regimen as safe according to protocol in all three patients, three additional patients were included and treated with neoadjuvant COMBO ICB. When the primary endpoint was met in at least 5 of 6 arm B patients and neoadjuvant COMBO ICB proved tolerable, phase IIa accrual was opened, in which an extension cohort of twenty patients was treated with COMBO ICB. See Fig. 1a for the trial design. Primary endpoint of phase IIa was efficacy, based on primary tumor pathological response rate at time of surgery and its relation with MR-RECIST-based response assessed on MRI obtained shortly prior to surgery. When IMCISION commenced, another primary outcome measure was to compare the effect of neoadjuvant ICB on infiltrating immune cells (assessed by immunohistochemistry and molecular analyses) in hypoxic versus normoxic tumor regions (determined per HX4-PET scan) within the same patient. Due to persistent radiopharmaceutical production problems, however, this outcome became unattainable and was abandoned in a protocol amendment of March 2019. Secondary and translational outcome measures included 2-year toxicity and survival parameters, baseline and on-treatment tumor hypoxia and molecular and immunological correlates of response to ICB.

Patients underwent routine baseline tumor staging using clinical examination under general anesthesia, MR imaging, ultrasonography-guided fine needle aspiration of cervical lymph nodes and FDG-PET. Baseline peripheral blood mononuclear cells (PBMCs) were obtained. During the examination under general anesthesia, a primary tumor biopsy was taken and immediately formalin-fixed and paraffin-embedded (FFPE). In addition, the primary tumor's margin was marked with tattoo ink as to safeguard surgical margins in the case of clinical tumor shrinkage after neoadjuvant ICB. Thus, the extent of surgery was not downscaled in patients with clinical evidence of response to ICB. MR imaging was repeated at the end of week 4. An additional FDG-PET scan was made in week 4 if the patient provided additional consent. Surgery was performed in week 5-6 at the NKI by experienced head and neck surgeons. Surgery generally consisted of cervical lymph node dissection, tumor resection and free vascularized or pedicled flap reconstruction of the defect. If indicated (e.g., T4 or pN2b status), adjuvant radiotherapy was administered according to institutional guidelines. In the case of extranodal tumor extension or incomplete tumor resection, concomitant platinum-based chemoradiotherapy was indicated. Adjuvant treatment was not altered based

on a patient's response to neoadjuvant ICB meaning that patients with a pretreatment indication for adjuvant (chemo)radiotherapy received such treatment even if an ICB response left them with a pathological stage that did not warrant adjuvant (chemo)radiotherapy. Follow-up consisted of frequent clinical examination during the early postoperative period and outpatient clinical examination every 3 months thereafter. Imaging in the follow-up period was performed when indicated.

**Pathological response evaluation**. Pathological response was determined on H&E-stained, FFPE sections of primary tumor obtained during surgery, by an experienced head and neck pathologist (LS). First, the histologically identifiable tumor bed area was determined. The tumor bed area was defined as the area in the resected specimen taken up by viable tumor cells plus the areas taken up by necrosis, keratinous debris, scarring and fibrosis, and multinucleated giant cell reaction (i.e., where immune-related regression of previously existent tumor was assumed to have taken place)[34]. The proportion of viable tumor cells within the tumor bed area was subsequently quantified as a percentage. We observed that in particular salvaged patients, who had previously undergone (chemo)radiotherapy, were characterized by a low baseline viable tumor cell percentage within fields of fibrosis and sometimes necrosis. To prevent misclassification of patients with low pretreatment tumor cellularity within areas of fibrosis induced by previous (radiotherapy) treatment as pathological responders, we compared the percentage of viable tumor cells in the on-treatment tumor bed to the percentage of viable tumor cells in the baseline biopsy. The percentage change between the viable tumor cell percentage in the baseline biopsy and the residual viable tumor cell percentage within the on-treatment, surgically resected tumor bed area was quantified. Aiming at a full tumor specimen analysis for pathologic response evaluation, H&E-stained slides from the on-treatment biopsies taken for research purposes were also included for pathological response examination. Patients were classified in response categories as proposed by Tetzlaff et al.[53]: patients with both ≤10% residual viable tumor cell percentage in the resected tumor bed and 90-100% decrease in viable tumor cells from baseline to on-treatment had a major pathological response (MPR). Patients with both ≤ 50% residual viable tumor cells and 50-89% decrease in viable tumor cell percentage from baseline to on-treatment had a partial pathological response (PPR), and patients with any percentage of residual viable tumor cells but < 50% change in viable tumor cell percentage had no pathological response (NPR). Adding the percentage change criterion made the classification of pathological response more conservative. In all translational analyses, patients with primary tumor MPR are compared to patients without MPR (PPR + NPR).

To meet the primary endpoint of pathological efficacy, the three patients that did not undergo whole tumor resection did have biopsies taken at the pre-planned time of surgery: two of the primary tumor and one of a distant lymph node metastasis. As we observed that pathological response is not necessarily homogeneous throughout a resection specimen and biopsies do not allow for full pathological evaluation of the tumor bed, biopsies may not be representative for PR. To maintain a uniform PR evaluation, we considered the biopsies of these three patients as insufficient surrogates for full-specimen pathological efficacy evaluation and excluded them from efficacy analysis. However, they were investigated for the presence of viable tumor, and for signs of pathological ICB response (neovascularization, proliferative fibrosis, lymphocyte infiltration)[28]. These patients' primary tumor samples were included in translational analyses and were classified according to their clinical and radiological response: 1 likely MPR and 2 likely NPR.

Response in lymph node metastases was assessed separately by quantifying the percentage of tumor bed occupied by viable tumor cells in H&E-stained, FFPE sections of affected lymph nodes. As spontaneous necrosis is not uncommon in HNSCC lymph node metastases, necrosis in the absence of other signs of ICB response (fibrotic scar tissue formation, neovascularization, multinucleated giant cell abundance, aggregates of macrophages) was not sufficient to qualify as a treatment response. Lymph nodal response categories were defined identically to primary tumors (with the absence of the comparison to the baseline biopsy, which was not available for lymph nodes), yet were not used to classify patients in one of the three response categories: achieving MPR in one or more affected lymph node sites in the absence of MPR in the primary tumor was not sufficient to be classified as a major pathological responder.

**Radiological and metabolic response evaluation**. Radiological response to neoadjuvant ICB was assessed in week 4 based on MR-RECIST version 1.1[29] by a head and neck radiologist (BJ) who was blinded to treatment regimen and outcome. FDG-PET scans at baseline and on-treatment were evaluated by a similarly blinded nuclear physician (WV) using Osirix (v 11.0.1, Pixmeo, Switzerland) for SUV$_{max}$ (signal intensity of most avid voxel), SUV$_{mean}$ (mean intensity of voxels within tumor volume with intensity ≥50% of SUV$_{max}$) and metabolic tumor volume (MTV, volume taken up by voxels within tumor with intensity ≥50% of SUV$_{max}$). Total lesion glycolysis (TLG) was defined as the product of MTV and SUV$_{mean}$. The 50% SUV$_{mean}$ of the tumor at baseline was used to define on-treatment MTV and TLG. MTV and TLG could not be reliably calculated in patients where the primary tumor could not be clearly visualized or accurately

distinguished from surrounding FDG-avid tissues. Change in TLG from baseline to on-treatment was calculated in percentages.

**Historical cohort formation**. After approval of the NKI institutional review board (file number IRBd20-106), a retrospective cohort of patients that underwent combined-approach (salvage) surgery for HNSCC without neoadjuvant treatment between January 2013 and December 2017 was composed and data were extracted from patients' files. All patients met IMCISION's main in- and exclusion criteria.

**Sample size calculation and statistical considerations**. Phase Ib was set up using a 3 + 3 design for both neoadjuvant regimens. For calculation of the phase IIa extension cohort size, the pathological response rate to neoadjuvant ICB was hypothesized at 33%. An incidence of <10% pathological response was considered clinically irrelevant. Including 26 patients treated with an identical (NIVO MONO or COMBO) ICB regimen and assuming a 33% pathological response rate meant an actual <10% response rate could be rejected with 90% power and 95% (one-sided) confidence.

Comparisons of continuous variables across ICB response categories were performed using a Wilcoxon rank-sum test. Pairwise comparisons of continuous variables within the same patient were performed using a Wilcoxon signed rank test. Time to progression (TTP) was defined as the time from surgery to first relapse event (local, regional or distant). A death without a relapse event was thus censored in the TTP analysis. The three patients that did not undergo surgery were excluded from TTP analysis, since they were never disease-free. Two overall survival (OS) analyses were performed: one from the date of surgery (concerning the 29 pathologically evaluable patients that actually underwent surgery) and one from the date of the first ICB dose (concerning all 32 patients). Death from any cause was defined an event in the OS analyses. Kaplan–Meier survival estimates were compared using a log-rank test. The reverse Kaplan–Meier method was used to calculate median follow-up time. Multiple logistic regression was performed to combine proposed biomarkers and ROC-analysis with area under the curve-determination was performed to assess its utility as a predictor for MPR. Analyses were performed in R (clinical data in v 4.0.3, multiplex and sequencing data in v 3.6.3) and Graphpad Prism (v.8.4.3) and all tests were two-sided. GSEA and differential gene expression analyses were corrected for FDR, all other analyses were not corrected for multiple testing. Due to the scarcity of the patient material, all translational investigations were performed once.

**Study oversight**. This study was an investigator-initiated trial with the NKI as sponsor. The NKI designed the study, collected and analyzed data, and wrote the manuscript. Funding was provided by Bristol-Myers Squibb through the International Immuno-Oncology Network and by the Riki Foundation. The trial protocol and its amendments were reviewed and approved by the Medical Research Ethics Committee of the Netherlands Cancer Institute—Antoni van Leeuwenhoek Hospital (MREC AVL, https://english.ccmo.nl/mrecs/accredited-mrecs/mrec-netherlands-cancer-institute-the-antoni-van-leeuwenhoek-hospital), under file number NL57794.031.16. The study's design and conduct were in accordance with all relevant regulations regarding the use of human study participants and the 1964 Helsinki declaration, and was consistent with Good Clinical Practice guidelines as formulated by the International Conference on Harmonization. All patients provided written informed consent prior to enrollment. The authors affirm that the patient of whom clinical photography is shown in Fig. 1c provided additional informed consent for publication of the images. The patient depicted in Supplementary Fig. 2 had died at time of writing, and additional informed consent for the publication of the photographs was obtained from the patient's partner.

**Immunohistochemistry**. HIF-1α, PD-L1, p16, p53, MLH1, MSH2, MSH6, PMS2, HCA2, HC10, β2M-, and ERG immunohistochemistry of FFPE primary tumor samples was performed on a BenchMark Ultra autostainer (Ventana Medical Systems). Briefly, paraffin sections were cut at 3μm, heated at 75 °C for 28 min and deparaffinized in the instrument with EZ prep solution (Ventana Medical Systems). Heat-induced antigen retrieval was carried out using Cell Conditioning 1 (CC1, Ventana Medical Systems) for 32 min (P16, P53, MLH1, MSH2, MSH6, HC10 and B2M), 48 min (PD-L1), 64 min (HIF-1α, HCA2), or 72 min (PMS2) at 95 °C, or for 28 min at 75 °C (ERG).

HIF-1α was detected using clone 54/HIF-1a (1/50 dilution, 64 min at 360 C, BD Transduction Laboratories, CatNo 610959), PD-L1 using clone 22C3 (1/40 dilution, 1 h at RT, Agilent / DAKO, CatNo M3653), p16 using clone MX007 (1/400 dilution, 32 min at 37 °C, ImmunoLogic, CatNo ILM 0632 C01), p53 using clone DO-7 (1/7000 dilution, 32 min at 37 °C, Agilent / DAKO, CatNo M7001), MLH1 using clone ES05 (1/20 dilution, 32 min at 37 °C, Agilent / DAKO, CatNo M3640), MSH2 using clone G219-1129 (Ready-to-Use, 12 min at 37 °C, Roche / Ventana, CatNo 8033684001), MSH6 clone EP49 (1/50 dilution, 32 min at 37 °C, Epitomics, CatNo AC-0047d), PMS2 using clone A16-4 (Ready-to-Use, 32 min at 37 °C, Roche / Ventana, CatNo 8033692001), HCA2 using a mouse monoclonal (1/2000 dilution, 1 h at RT, Nordic Mubio, CatNo MUB0236P), HC10 using a mouse monoclonal (1/20000 dilution, 32 min at 37 °C, Nordic Mubio, MUB2037P), β2M using a polyclonal (1/4500 dilution, 32 min at 37 °C, DAKO / Agilent,

CatNo A0072) and ERG using clone EPR3864 (Ready-to-Use, 16 min at 37 °C, Roche Diagnostics, CatNo 6478450001).

For MLH1 and PMS2, signal amplification was applied using the Optiview Amplification Kit (4 min, Ventana Medical Systems). Bound antibody was visualized using the OptiView DAB Detection Kit (Ventana Medical Systems). Slides were counterstained with Hematoxylin II and Bluing Reagent (Ventana Medical Systems). B2M-bound antibody was visualized using OmniMap anti-Rabbit HRP (Ventana Medical systems) for 12 min, followed by the ChromoMap DAB detection kit (Ventana Medical Systems).

Scoring was performed by two head and neck pathologists (HIF-1α by SW, HLA, microsatellite stability and CPS by LS). HIF-1α was scored by determining the percentage of tumor cells that express HIF-1α. In two patients with MPR, no residual tumor was left after ICB and their on-treatment HIF-1α score was scored '0%'. PD-L1 expression on tumor cells, macrophages and lymphocytes was quantified divided by the total number of viable tumor cells to calculate the combined positive score (CPS) as a percentage[9]. p16 was scored as positive or negative and p53 staining was scored as negative, wild-type, or overexpressed. Positivity for p16 with a wild-type p53 expression was considered as HPV-positivity. HPV-positivity by IHC was validated on the molecular level with PCR. MLH1, MSH2, MSH6, PMS2 were scored as positive or negative: negativity for any staining was considered microsatellite instability. HCA2, HC10 and β2M were scored by assigning a score of 0-5 for the percentage of positive tumor cells (0 = <1%, 1 = 1-5%, 2 = 6-25%, 3 = 26-50%, 4 = 51-75%, 5 = >75%) summed up with a score of 1-3 for staining intensity (1 = absent, 2 = weak, 3 = strong) was assigned to each staining. For HCA2 and HC10, a score of 1 was considered negative, 2-4 weak, 5-6 moderate and 7-8 high expression. β2M scoring was dichotomized between negative (1-4) or positive (5-8)[54]. Patients were considered HLA class 1-proficient if they scored 'positive' for β2M or at least 'weak' for HCA2 or HC10.

Microvessel density (MVD) was assessed by two researchers (JV and LS) by staining endothelial cells using ERG immunohistochemistry. The region of highest neovascularization in the tumor-associated stroma, defined as stroma between tumor fields and peritumoral stroma no further than 0.5 mm from the nearest tumor cell, was identified by scanning at low power (×40). Any clearly separable brown-staining cluster of at least two endothelial cells was counted as a vessel: individual cells were not counted. An identifiable vessel lumen was not necessary for a vessel to be counted. Thick-walled arterioles were not counted[55].

**Multiplex immunofluorescence**. Paraffin sections were cut at 3 µm, dried overnight and stored at +4 °C. Slides were heated for 30 min at 70 °C. Staining was performed on a Ventana Discovery Ultra automated stainer, using the Opal 7-Color Manual IHC Kit (50 slides kit, Perkin Elmer, CatNo NEL81101KT). Protocol starts with heating for 28 min at 75 °C, followed by deparaffinizing with Discovery Wash using the standard setting of three cycles of 8 min at 69 °C. Pre-treatment was performed with Discovery CC1 buffer for 32 min at 95 °C, after which Discovery Inhibitor was applied for 8 min to block endogenous peroxidase activity. Specific markers were detected consecutively on the same slide with the following antibodies: anti-CD3 (Clone SP7, 1/400 dilution, 1 h at RT, Thermo-Scientific, CatNo RM-9107-S), anti-CD8 (Clone C8/144B, 1/100 dilution, 1 h at RT, DAKO, CatNo M7103), anti-CD68 (Clone KP1, 1/500 dilution, 1 h at RT, DAKO, CatNo M0814), anti-FoxP3 (clone 236 A/47, 1/50 dilution, 2 h at RT, Abcam, CatNo ab20034), Anti-CD20 (Clone L26, 1/500 dilution, 1 h at RT, DAKO, CatNo M0755), Anti-PanCK (Clone AE1AE3, 1/100 dilution, 2 h at RT, ThermoScientific, CatNo MS-343P). Each staining cycle consisted of four steps: primary antibody incubation, OPAL polymer HRP Ms+Rb secondary antibody incubation (Ready-to-use, 32 min at RT, PerkinElmer, CatNo ARH1001EA), OPAL dye incubation (OPAL520, OPAL540, OPAL570, OPAL620, OPAL650, OPAL690, 1/50 or 1/75 dilution as appropriate for 32 min at RT) and an antibody denaturation step using CC2 buffer for 20 min at 95 °C. Cycles were repeated for each new antibody to be stained. At the end of the protocol slides were incubated with DAPI (1/25 dilution in Reaction Buffer) for 12 min. After the run was finished, slides were washed with demineralized water and mounted with Fluoromount-G (Southern Biotech, cat 0100-01) mounting medium.

Slides were imaged using the Vectra 3.0 automated imaging system (PerkinElmer). First, whole slide scans were made at 10x magnification. Then, multispectral images were taken at 20x magnification. Library slides were created by staining a representative sample with each of the specific dyes. Using InForm software version 2.4 and the library slides, the multispectral images were unmixed into eight channels: DAPI, OPAL520, OPAL540, OPAL570, OPAL620, OPAL650, OPAL690 and Auto Fluorescence and exported to a multilayered TIFF file.

The multilayered TIFF's were fused with HALO software (v3.0) to create one file for each sample. Tissue annotation and cell identification was performed by three researchers (JV, IS, and LS), one of whom is an experienced head and neck pathologist. A Random Forest classifier was used to make tissue classifications for tumor and stroma class with resolution 2 (µm/px) and minimum object size 200 (µm²). We did not define regions of interest: annotations of tumor and stroma were generated via the tissue classifier on whole slides and adjusted where necessary. Salivary glands, if present, were annotated manually. Intratumoral infiltrate was defined as the presence of immune cells located within the tumor annotation layer. The Indica Labs Highplex FL v3.0.3 algorithm was used for analysis. All annotation

layers were analyzed and both the summary data and object data were exported in comma separated value files using the export manager in HALO.

Cell phenotype quantification was done in R (V3.6.3) based on the HALO object files containing cell marker positivity, cell coordinates, and tissue type (tumor, stroma, or salivary gland). Cell phenotypes were quantified per tissue type (tumor, stroma, or salivary gland) in whole slides and normalized for tissue region surface area in mm².

**DNA and RNA sequencing.** Tumor DNA and RNA was isolated from formalin-fixed paraffin-embedded (FFPE) primary tumor sections containing at least 30% viable tumor cells, except for on-treatment samples with a complete pathological response and 1 salvaged patient's sample that was characterized by 20% viable tumor cell count at baseline. A pathologist (LS) scored the tumor percentage and indicated the most tumor-dense region on a hematoxylin and eosin (H&E) stain slide for subsequent DNA/RNA isolation. Five to 10 FFPE slides (10 μm) were used for simultaneous isolation of DNA and RNA using the AllPrep DNA/RNA FFPE isolation kit (Qiagen, 80234) and the QIAcube, according to the manufacturer's protocol. Germline DNA was isolated from PBMCs using AllPrep DNA / RNA / miRNA Universal isolation kit (Qiagen, 80224) and the QIAcube, according to the manufacturer's protocol. Both whole-exome and RNA sequencing were performed by CeGaT.

DNA exome sequencing libraries were generated using 50 ng of DNA with the Twist Human Core Exome Plus (Twist Bioscience). Libraries were sequenced on a NovaSeq 6000 using 2×100bp to an average of 30× for tumor samples and 100× for blood samples. Demultiplexing of the sequencing reads was performed with Illumina bcl2fastq (2.20). Adapters were trimmed with Skewer (v 0.2.2)[56]. The quality of FASTQ files was analyzed with FastQC (version 0.11.5-cegat)[57]. Sequence reads were mapped with BWA (0.7.12) to the human reference genome GRCh38. Downstream data pre-processing was performed using the GATK4 (v 4.0.6.0) workflow for variant calling[58]. The pre-processed data were used for variant calling using GATK4 Mutect2[58]. All identified mutations were required to have passed all Mutect2 tests (FILTER field equals 'PASS'). Variants were subsequently annotated using VEP[59]. Tumor mutational burden (TMB) was calculated by summarizing the total number of nonsynonymous, somatic mutations per sample with minimal variant allele frequency (VAF) of 0.02 (2%). The COSMIC mutational signatures (V2 March 2015)[36] were assessed using MutationalPatterns (v. 1.12.0)[60]. Both nonsynonymous and synonymous somatic mutations with a minimal VAF of 0.02 were used to calculate the relative contribution of each of the 30 COSMIC signatures in each sample.

RNA sequencing libraries were generated using 10 ng of RNA with the SMART Stranded Total RNA Seq Kit (Takara). Libraries were sequenced on a NovaSeq 6000 using 2×100bp to an average of 50 million reads pairs per sample. Demultiplexing of the sequencing reads was performed with Illumina bcl2fastq (2.20). Adapters were trimmed with Skewer (v 0.2.2)[56]. the first three nucleotides of the second sequencing read (Read 2) are derived from the Pico v2 SMART Adapter. Those three nucleotides have been trimmed with cutadapt (v 1.12)[61]. The quality of FASTQ files was analyzed with FastQC (version 0.11.5-cegat)[57]. Fastq files were mapped to the human reference genome (Homo.sapiens.GRCh38.v82) using STAR(v 2.6.0c)[62] in 2-pass mode with default settings. Count data generated with HTseq-count[63] was analyzed with DESeq2 (v 1.26.0)[64]. Centering of the normalized gene expression data per dataset was performed by subtracting the row means and scaling by dividing the columns by the standard deviation.

**Gene signatures.** The IFNγ[65] score was defined as the average expression (based on the Z-Score) of 28 genes. The joint chronic hypoxia score[33] was defined as the average expression (based on the Z-Score) of 99 genes. To quantify abundance of immune and stromal cell populations we applied the MCPcounter[66] and Danaher genesets[67].

**Geneset enrichment analysis.** PreRanked GSEA was based on gene lists ranked on the Signal2Noise ratio and performed using the BROAD javaGSEA standalone version (http://www.broadinstitute.org/gsea/downloads.jsp) and the curated 'hallmark genesets' (http://software.broadinstitute.org/gsea/msigdb/collections.jsp). Genesets with an FDR < 0.1 were considered significant.

**Reporting summary.** Further information on research design is available in the Nature Research Reporting Summary linked to this article.

## Data availability

The raw DNA and RNA sequencing data generated in this study have been deposited in the European Genome-phenome Archive (EGA) under accession codes EGAS00001005466 and EGAS00001005454. Sequencing reads were mapped to the human genome [Homo.sapiens.GRCh38.v82, https://www.ncbi.nlm.nih.gov/assembly/GCF_000001405.26/]. The previously published IFNγ[65], joint chronic hypoxia[33], MCPcounter[66] and Danaher[67] genesets were used in this paper. PreRanked geneset enrichment analysis (GSEA) was performed using the BROAD javaGSEA standalone version (http://www.broadinstitute.org/gsea/downloads.jsp) and the curated 'hallmark genesets' (http://software.broadinstitute.org/gsea/msigdb/collections.jsp). DNA, RNA, multiplex immunofluorescence and all other relevant, de-identified clinical data of individual patients are available under restricted access. Access to these data can be

obtained upon scientifically sound request with the NKI's scientific repository at repository@nki.nl, who will contact corresponding author C.L.Z.. All requests will be reviewed by the institutional review board (IRB) of the Netherlands Cancer Institute (NKI), and will require the requesting researcher to sign a data access agreement with the NKI. Source data are provided with this paper, from which all processed and annotated data (FDG-PET, immunohistochemistry, multiplex immunofluorescence, processed DNA and RNAseq) that underlie the Figs. (1b, 1e-g, 2a-c, 2e-h) and Supplementary Figs. (5d, 6a-g, 7, 8a-b) are available. The most recent version of the Trial Protocol is available under Supplementary Note 1 from the Supplementary Information file. Source data are provided with this paper.

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

## Acknowledgements

We thank all patients and their families for their participation in the study. We thank our colleagues from the NKI Core Facility Molecular Pathology & Biobanking (D. Peters, S. Cornelissen, W. Hoefakker, M. Alkemade, L. Borghuis, and I. Hofland) for supplying lab support; the Radiology & Nuclear Medicine department (A. Paape, A. van Raamsdonk, and C. Vroonland) and the OR personnel (R. van Rossum, J. Hogenboom, J. Peerenboom, and S. Hermans) for their support in planning the various trial procedures; the NKI Laboratory department (T. Korse, E. Platte, and M. Lucas) for PBMC acquisition and isolation; the NKI scientific administration (A. Keijser, L. Ruiter, and E. van Schaffelaar) for data management; the department of Medical Oncology (M. Tesselaar, H. van Thienen, S. Wilgenhof, and M. Chalabi) and the Head and Neck Surgery and Oncology nurse practitioners (H. Tefsen and M. Kroon) and residents for (outpatient) clinical care; the members of the Tumor Biology & Immunology and Molecular Oncology & Immunology departments for valuable discussions; S. Vanhoutvin for financial management and legal support; and Bristol-Myers Squibb for scientific input and financial support. This study was funded by the BMS International Immuno-Oncology Network and the Riki Foundation, while the NKI was the study's sponsor. Both funding sources had no role in design and execution of the study, data analysis or writing of the manuscript.

## Author contributions

J.L.V. and J.B.W. coordinated trial procedures, were responsible for outpatient care, analyzed and interpreted clinical and translational data, and wrote the manuscript. C.L.Z. designed the study, oversaw trial procedures, performed surgeries, and wrote the

manuscript. O.K. and J.J.H.T. performed molecular bioinformatics analyses. X.Q. and A.M.vd.L. provided scientific input during trial design and manuscript writing. Y.L. performed multiplex IF bioinformatics analyses. I.M.S. created multiplex IF cell segmentation algorithms and provided lab support. L.A.S. performed pathological response evaluation, IHC and IF scoring and scientific support during trial design and manuscript writing. S.M.W. performed IHC scoring. M.W.M.vd.B., R.D., M.B.K., L.K., W.M.C.K., P.J.F.M.L., W.H.S., L.E.S., and L.vd.V. performed surgeries, provided scientific input during trial design, and were responsible for postsurgical clinical trial patient care. I.B.T. referred patients and provided scientific input during trial design. S.O. was responsible for (outpatient) clinical care of trial patients. J.P.d.B. was responsible (for outpatient) clinical care and performed scientific input during trial design. B.J. performed radiological analyses. W.V.V. performed nuclear imaging analyses. A.A.M. oversaw postoperative radiotherapy and provided scientific input during trial design. A.K. was the clinical data manager. V.vd.N. performed statistical analyses relevant to trial design and clinical data. A.B. supervised translational lab work and was responsible for storing and processing of tumor samples. E.H. supervised multiplex IF. D.S.P. supervised bioinformatics analyses. T.N.M., C.U.B., J.B.A.G.H., V.vd.N., and C.L.Z. designed the trial and made the experimental plan of action.

## Competing interests

J.L.V., J.B.W.E., J.J.H.T., X.Q., A.vd.L., Y.L., I.S., L.S., S.O., B.J., W.V.V., A.A.M., V.vd.N., A.K., E.H., A.B., R.D., L.K., M.B.K., P.J.F.M.L., W.H.S., W.M.C.K., L.vd.V., and I.B.T. declare no competing interests. C.L.Z. reports receiving institutional research financial support from BMS to fund the present trial. M.W.M.vd.B. reports, outside the submitted work, institutional research funding from ATOS Medical. S.M.W. reports, all outside the submitted work: institutional research funding from Roche, Pfizer, MSD, Bayer, Amgen, BMS, AstraZeneca, Lilly and Nextcure. O.K. is employed at Neogene Therapeutics B.V., though at time of analysis and writing his employment was at the NKI. O.K. further reports, outside the submitted work, a pending patent application (title: "Gene signatures and method for predicting response to pd-1 antagonists and ctla-4 antagonists, and combination thereof", number: WO2020005068A8). D.S.P. reports, all outside the submitted work, a pending patent application (title: "Gene signatures and method for predicting response to pd-1 antagonists and ctla-4 antagonists, and combination thereof", number: WO2020005068A8) and a role as co-founder, shareholder and advisor of Immagene. J.P.d.B. reports, all outside the submitted work: institutional research funding from Merck KGaA; institutional honoraria for an advisory role for MSD. T.N.M.S. reports, all outside the submitted work: advisory roles for Adaptive Biotechnologies, AIMM Therapeutics, Allogene Therapeutics, Merus, Neogene Therapeutics, Neon Therapeutics and Scenic Biotech; research support from Merck KGaA; stockholdership of AIMM Therapeutics, Allogene Therapeutics, Merus, BioNTech, Neogene Therapeutics, and Scenic Biotech. C.U.B. reports, all outside the submitted work: institutional research funding from BMS, Novartis and Nanostring; institutional honoraria for advisory roles for BMS, MSD, Roche, Novartis, GSK, AZ, Pfizer, Lilly, Genmab and Pierre Fabre; personal honoraria for an advisory role for Third Rock Ventures; stock ownership of Uniti Cars and Immagene. J.B.A.G.H. reports, all outside the submitted work: institutional honoraria for advisory roles for AIMM, Amgen, BioNTech, BMS, GSK, Ipsen, MSD, Merck Serono, Molecular Partners, Neogene Therapeutics, Novartis, Pfizer, Roche/Genentech, Sanofi, Seattle Genetics, Third Rock Ventures, Vaximm; stock option ownership of Neogene Therapeutics; Institutional research funding from Amgen, BioNTech, BMS, MSD, Novartis.

## Additional information

[1]Department of Head and Neck Surgery and Oncology, The Netherlands Cancer Institute, Amsterdam, The Netherlands. [2]Department of Radiation Oncology, Erasmus University Medical Center, Rotterdam, The Netherlands. [3]Neogene Therapeutics, Amsterdam, The Netherlands. [4]Division of Molecular Oncology & Immunology, The Netherlands Cancer Institute, Amsterdam, The Netherlands. [5]Division of Tumor Biology & Immunology, The Netherlands Cancer Institute, Amsterdam, The Netherlands. [6]Department of Pathology, The Netherlands Cancer Institute, Amsterdam, The Netherlands. [7]Department of Pathology and Medical Biology, Groningen University Medical Center, Groningen, The Netherlands. [8]Department of Oral and Maxillofacial Surgery, Amsterdam University Medical Center location AMC, Amsterdam, The Netherlands. [9]Department of Otorhinolaryngology Head and Neck Surgery, Maastricht University Medical Center+, Maastricht, The Netherlands. [10]Department of Radiology, Amsterdam University Medical Center location VUmc, Amsterdam, The Netherlands. [11]Department of Nuclear Medicine, The Netherlands Cancer Institute, Amsterdam, The Netherlands. [12]Department of Radiation Oncology, The Netherlands Cancer Institute, Amsterdam, The Netherlands. [13]Department of Biometrics, The Netherlands Cancer Institute, Amsterdam, The Netherlands. [14]Core Facility Molecular Pathology & Biobanking, The Netherlands Cancer Institute, Amsterdam, The Netherlands. [15]Oncode Institute, Utrecht, The Netherlands. [16]Department of Medical Oncology, The Netherlands Cancer Institute, Amsterdam, The Netherlands. [17]Department of Otorhinolaryngology Head and Neck Surgery, Leiden University Medical Center, Leiden, The Netherlands. [18]These authors contributed equally: Joris L. Vos, Joris B. W. Elbers. ✉email: c.zuur@nki.nl

