## [Peer Review File · Nature Communications]

Reviewers' Comments:

Reviewer #1:

Remarks to the Author:

The authors have revised their manuscript helpfully with a number of clarifications and improvements to the manuscript. Some minor revisions would add clarity for the reader as below. For instance they could further detail the Clavien-Dindo classification of complications for mono versus combo therapy. In addition although there were only six nivolumab treated patients given the difference in response rate it would seem that these should not be grouped in with the combination arm.

They should comment on whether there was any correlation between radiographic response using MRI and pathologic response

Their response and rebuttal point #4 indicates that adjuvant therapy was not altered based on the response extent, however this raises a question and some confusion. Since pathologic staging drives the selection of adjuvant therapy, and if as they say residual disease was seen in very few or recent one out of 29 patients, were adjuvant treatment decisions based on the pre-operative scans, since the number of positive lymph nodes and or extra Nodal extension can be very difficult to tell preoperatively. Next because they state that microscopic residual disease after surgery was present in only one of the 29 patients, it is unclear whether they are adding quantitatively residual tumor from the pathologic specimen subjected for standard of care analysis at surgery as well as residual disease in the research specimen/biopsy from surgery? Otherwise the research biopsy should be added to these data regarding the presence of microscopic residual disease which was an analysis that was only performed on the surgical specimen when a biopsy had already been taken from it.

Reviewer #2:

Remarks to the Author:

One specific comment on the analyses presented in Extended Data Figure 8.

Lines 362 and 365 and Extended Data Fig 8. It is not clear statistical test has been used to get the p-values presented on lines 1185 and 1189. The multiple regression analyses done to develop a combination of the two biomarkers related to Extended Data Fig 8a and Fig 8b given the small sample size with only 9 MPR may be subject to overfitting and result in an optimistic AUC. One way to check this is to estimate optimism using bootstrapping.

Reviewer #3:

Remarks to the Author:

This work is original and highly significant given the morbidity of current treatments for this disease. The manuscript is clearly written and presents a wide range of correlative studies. The authors provided the requested clarification of the methodology for assessing pathologic response and addressed all major and minor points of my first review to my satisfaction. I support the publication of this manuscript.

One minor suggestion is to add to the methods the number of multiplex IF regions of interest analyzed per patient and how the ROIs were selected (e.g., to include tumor, stroma, or both; any preference given to regions of inflammation?).

Following Dr. Danelli and the Nature Communications Editorial Team's suggestion, we have updated our postsurgical follow-up, which now amounts to a median of 24 months (see page 3, lines 73-74 and page 9, line 233-235). All survival analyses and figures were updated correspondingly. In addition, we have separated the NIVO MONO from the COMBO ICB treatment groups when reporting on irAEs, surgical outcomes and complications, response rates, and survival estimates. Finally, we have formulated a point-by-point response to the three reviewers' remaining comments below.

Reviewer #1

The authors have revised their manuscript helpfully with a number of clarifications and improvements to the manuscript. Some minor revisions would add clarity for the reader as below.

- 1. For instance they could further detail the clavien dindo classification of complications for mono versus combo therapy.**

We thank you again for carefully reviewing our manuscript. We have modified Supplementary Table 3 (Supplementary Information file, page 5), which now shows the surgical variables and complications for the NIVO MONO and COMBO groups separately.

- 2. In addition although there were only six nivolumab treated patients given the difference in response rate it would seem that these should not be grouped in with the combination arm.**

Our trial was indeed not designed or powered to investigate any potential difference between nivolumab monotherapy (NIVO MONO) and nivolumab + ipilimumab combination therapy (COMBO). Following your remarks, we have ungrouped the NIVO MONO and COMBO cohorts throughout the manuscript when reporting irAEs (see Supplementary Table 2, supplementary information file page 4 and Manuscript page 7, lines 173-188), surgical variables and complications (see Supplementary Table 3, supplementary information file page 5 and Manuscript page 7, lines 189-195), response rates after neoadjuvant ICB (see Figure 1d on page 37 and page 8-9 lines 209-233), and survival (see page 9-10 lines 234-269). We have also

modified our conclusion accordingly (see page 17, lines 463-475).

3. They should comment on whether there was any correlation between radiographic response using MRI and pathologic response.

Following your comment, we have included Supplementary Table 6 to the Supplementary Information (on page 8 of Supplementary Information File), showing a matrix of the 24 patients evaluable per MR-RECIST who also had pathological response evaluation available. We show that MR-RECIST has a 100% specificity but only a 29% sensitivity to detect MPR after neoadjuvant ICB but prior to surgery (Supplementary Table 6, Supplementary Information file page 8 and Manuscript file page 10, lines 280-283).

4. Their response and rebuttal point #4 indicates that adjuvant therapy was not altered based on the response extent, however this raises a question and some confusion. Since pathologic staging drives the selection of adjuvant therapy, and if as they say residual disease was seen in very few or recent one out of 29 patients, were adjuvant treatment decisions based on the pre-operative scans, since the number of positive lymph nodes and or extra Nodal extension can be very difficult to tell preoperatively.

We thank you for your inquiry. We fully agree that pathological staging drives the indication for adjuvant therapy in HNSCC. In IMCISION, the decision to treat with adjuvant (chemo)radiotherapy was based on pre-operative clinical staging (including imaging tumor characteristics and physical examination), as well as pathologic staging at time of surgery. By stating that we did not alter adjuvant treatment based on response to immunotherapy, we mean that patients with a pre-treatment indication for postoperative radiotherapy (e.g. T4 status) were still treated with adjuvant radiotherapy (even though the remaining tumor after neoadjuvant immunotherapy may have been only pT1 or pT2). We have added this clarification on page 20, lines 539-541. However, we indeed cannot rule out the possibility that some MPR patients without an indication for adjuvant (chemo)radiotherapy based on clinical staging were cleared of potential pathological indications for postoperative (chemo)radiotherapy by their deep response to immunotherapy. If anything, however, this may have led to a relative adjuvant undertreatment of patients with an MPR, making it unlikely that this group's superior survival was caused by adjuvant treatment.

Following your comment, we have added these deliberations to the Discussion (page 14, lines 386-394).

- 5. Next because they state that microscopic residual disease after surgery was present in only one of the 29 patients, it is unclear whether they are adding quantitatively residual tumor from the pathologic specimen subjected for standard of care analysis at surgery as well as residual disease in the research specimen/biopsy from surgery? Otherwise the research biopsy should be added to these data regarding the presence of microscopic residual disease which was an analysis that was only performed on the surgical specimen when a biopsy had already been taken from it.**

We fully agree that the both the surgical specimen and the biopsies taken for research purposes should be included in the determination of residual viable tumor. To facilitate this, our trial protocol made sure that an H&E-stained side was available for all research biopsies (see page 50 of the Trial Protocol), which were taken along in the pathological response assessment. We have added this statement to the Methods section on page 20, lines 560-562.

Reviewer #2

One specific comment on the analyses presented in Extended Data Figure 8.

1. **Lines 362 and 365 and Extended Data Fig 8. It is not clear statistical test has been used to get the p-values presented on lines 1185 and 1189. The multiple regression analyses done to develop a combination of the two biomarkers related to Extended Data Fig 8a and Fig 8b given the small sample size with only 9 MPR may be subject to overfitting and result in an optimistic AUC. One way to check this is to estimate optimism using bootstrapping.**

We thank you again for your review of our manuscript. We agree that an ROC with AUC-determination on a sample with only 9 MPR-events may yield too optimistic results and could lead to drawing inappropriate conclusions from our data. After reconsideration, we have decided to simply show the bubble plots in Extended Data Fig. 8 and to omit ROC-analysis and corresponding AUC with P-values from our manuscript (see page 13, lines 358-366 and the Extended Data Figure 8 legend on page 51).

Reviewer 3

This work is original and highly significant given the morbidity of current treatments for this disease. The manuscript is clearly written and presents a wide range of correlative studies. The authors provided the requested clarification of the methodology for assessing pathologic response and addressed all major and minor points of my first review to my satisfaction. I support the publication of this manuscript.

1. **One minor suggestion is to add to the methods the number of multiplex IF regions of interest analyzed per patient and how the ROIs were selected (e.g., to include tumor, stroma, or both; any preference given to regions of inflammation?).**

We thank you again for your careful review and compliments on the work. To prevent bias in the present work, we did not make use of any regions of interest, but rather annotated tumor regions, stromal regions and salivary glands on the whole slides. In the manuscript, we report actual intratumoral infiltration (i.e. located within the tumor annotation layer) of immune cells before and after neoadjuvant treatment based on the whole slides, and normalize for tissue surface area in mm^2 . We now stress this on page 26, in lines 721-730.

Reviewers' Comments:

Reviewer #1:

Remarks to the Author:
suitably revised

Reviewer #2:

Remarks to the Author:
The responses addressed the issue in my previous review.

Reviewer #3:

Remarks to the Author:
I appreciate the clarification on the multiplex IF sampling. All of my concerns have been addressed.